# Mental health concerns and needs of international students in higher education settings: A scoping review protocol

Kira Rudakova[1,2], Shalini Lal[1,2,3]*

1 School of Rehabilitation, Université de Montréal, Montréal, Québec, Canada, 2 Youth Mental Health and Technology Lab, Research Centre, Centre Hospitalier de l'Université de Montréal, Montréal, Québec, Canada, 3 Douglas Research Centre, Montréal, Québec, Canada

* shalini.lal@umontreal.ca

## Abstract

### Introduction

The transition to higher education is a significant milestone for many individuals; however, it also brings new stressors and challenges, particularly for international students adjusting to life in a foreign country. Despite the increasing diversity of student populations globally, there remains a gap in existing reviews that capture the full scope of international student mental health concerns and needs. Existing reviews on the mental health and psychosocial adjustment of international students often concentrate on acculturation stress, thus overlooking other mental health concerns such as depression and anxiety, as well as positive mental health experiences like increased well-being. Meanwhile, other reviews tend to focus more heavily on specific regions, such as the United States and Australia, or student populations, particularly East-Asian students. While valuable, this focus may limit our understanding of the diverse mental health experiences of international students.

### Objective

The objective of this scoping review is to map and summarize research on international students' mental health experiences and concerns (e.g., depression) as well as factors influencing their well-being (e.g., social support and institutional resources).

### Methods

A search strategy guided by the Joanna Briggs Institute (JBI) Manual of Evidence Synthesis has been developed according to the Population, Concern, and Context (PCC) framework and will be applied to four electronic databases (i.e., MEDLINE, Embase, PsycInfo, and CINAHL). Two reviewers will pilot the selection strategy on subsets of 10 articles until a 90% agreement rate is achieved. Once this rate is

**Data availability statement:** No datasets were generated or analysed during the current study. All relevant data from this study will be made available upon study completion.

**Funding:** This research was supported, in part, by the Fonds de recherche du Québec –Santé (FRQS) (to KR) and the Faculty of Medicine at the Université de Montréal (to KR) through academic scholarships. This research was also supported, in part, thanks to funding from the Canada Research Chairs Program (to SL) through a salary award and operational funds. The sponsors or funders had no role in the study design, data collection and analysis, decision to publish, or preparation of the manuscript.

**Competing interests:** We have read the journal's policy and the authors of this manuscript have the following competing interests: KR is a graduate student at the Université de Montréal and is conducting this work towards partial fulfillment of requirements for their Master's of Science degree under the supervision of SL. This does not alter our adherence to PLOS ONE policies on sharing data and materials.

reached, a single reviewer will screen the remaining articles independently. Two reviewers will pilot data extraction on subsets of 10 included studies, after which one reviewer will proceed independently. Main findings will be presented through descriptive statistics, using tables and figures.

## Expected contributions

This scoping review will assess existing literature on the mental health needs and experiences of international students, highlighting overlooked issues, such as challenges beyond acculturation stress and the experiences of underrepresented student populations, including those studying outside of Western countries. Ultimately, the findings may identify areas for further research and inform educational institutions and mental health professionals in developing support resources that can effectively address diverse needs of international students.

## Introduction

The transition to higher education (e.g., university and college) is a significant milestone, often marked by excitement, new opportunities, and personal growth [1–3]. At the same time, this transition introduces a range of academic, personal, and emotional challenges that can have a significant impact on student mental health and well-being. As they navigate increased academic and life changes, university students are burdened with heightened feelings of stress, anxiety, and depression [4]. These mental health challenges, in turn, impact their academic performance and persistence. As such, mental health in higher education has become a growing focus of research and policy efforts [5].

Although challenges of higher education are universal, international students face additional complexities. For instance, a review by Smith and Khawaja [6] on acculturative experiences of international students reported that being far from familiar support systems often means starting over a in a new country, which can be a difficulty and isolating process. When combined with language barriers and cultural differences, students may struggle to form new connections and adjust, leading to feelings of loneliness, homesickness, and emotional distress. These adjustment difficulties can manifest in symptoms of depression, anxiety, or other mental health concerns. The review [6] also highlighted that cultural differences in academic expectations and teaching styles may result in academic stress and performance anxiety among international students, especially if they feel pressure to excel in unfamiliar educational environments.

With these complexities in mind, it is not surprising that recent years have seen a significant rise in mental health concerns among the international student population. For example, a longitudinal study by King et al. [7] found that international students reported increased symptoms of depression and anxiety by the end of their first academic year abroad. Similarly, a multi-year study by Prado et al. [8] reported that international students experienced significantly higher levels of depression, stress,

and loneliness, alongside lower perceived social support and resilience. Together, these findings point to a consistent pattern of elevated psychological distress among international students, emphasizing the need for a better understanding of their mental health challenges.

Despite the rising prevalence of these mental health challenges among international students, there remains a scarcity of literature reviews that address these issues. Existing literature often focuses on specific aspects of international student experiences, such as acculturation, without providing an in-depth analysis of their mental health status and needs. For instance, reviews by Smith and Khawaja [6] and Alharbi and Smith [9] primarily discuss predictors of psychosocial adjustment but only briefly mention mental health outcomes, doing so from an acculturation standpoint (i.e., acculturation stress). Although acculturation can play an important role in psychosocial adjustment, it should not be assumed as the default reason for all mental health-related challenges faced by international students. Furthermore, some reviews [10,11] including a scoping review by McKenna et al. [12], which explores challenges faced by international students in health professions, focus primarily on the negative factors that impact student wellbeing. These reviews often overlook the positive aspects that can enhance mental health, such as social support, resilience, and effective coping strategies. Consequently, they may miss nuanced elements of international student mental health experiences.

In addition to the limited attention to positive mental health experiences and the broader landscape of student well-being, there is a noticeable trend in the literature where specific student populations and regions are more frequently studied. For instance, much of the existing reviews on international student mental health tends to concentrate on East-Asian students [13,14] and on countries like the United States and Australia [15,16]. A systematic review by Zhang and Goodson [13] exemplifies this trend, providing a comprehensive analysis of predictors for the psychosocial adjustment of international students in the United States, yet predominantly focusing on East Asian students. Similarly, Li et al. [14] conducted a systematic review offering a multifaceted insight into the psychological well-being of international students; however, their review also primarily centers on East Asian students. Meanwhile, a systematic review by Brunsting et al. [15] on the predictors of undergraduate international student psychosocial adjustments exclusively focuses on universities in the United States. While such reviews provide valuable insights into specific populations and contexts, they also underscore a need to expand the scope of research to more comprehensively reflect the diverse experiences of international students from a variety of cultural, regional, and institutional settings. For example, a qualitative investigation on assimilation of Syrian refugees studying in Turkey by Safak-Ayvazoglu et al. [16] revealed that despite benefiting from Arabic language and the safety of their host country, these students still face significant mental health challenges due to trauma from war. While certain aspects of their environment may facilitate adaptation, the enduring impact of their backgrounds and experiences can lead to complex mental health concerns. International students come from a variety of cultural backgrounds, each with its own set of values, norms, and challenges related to mental health. As a result, overlooking this diversity limits our understanding of the cross-cultural experiences of international students.

Furthermore, a holistic understanding of international student mental health requires consideration of the host country's cultural context. Different host countries have distinct educational systems, social climates, and support services available to international students, all of which can significantly impact their mental well-being [17]. By exclusively focusing on specific regions like the United States, researchers may fail to capture the unique stressors and resources present in other host countries, thereby limiting the applicability of their findings on a global scale. Therefore, there is a need for a review that encompasses a broader range of cultural backgrounds and host countries to gain a more nuanced understanding of international student mental health concerns, needs, and experiences. Through such research, common stressors and protective factors across cultures will be identified, and unique challenges faced by students in different contexts will be highlighted.

While achieving a truly comprehensive review covering all mental health experiences and cultural contexts remains to be a challenging task, the existing literature highlights key gaps that need attention. This underscores the importance of this scoping review, which aims to systematically map the existing literature on mental health among international

students. This will help to identify common themes and trends across studies, highlighting the most prevalent issues faced by international students in higher education settings. Through this, future research and policy makers can develop interventions and support services that are not only culturally sensitive but also effective in addressing diverse student needs.

Numerous factors informed the decision to opt for scoping review methodology. First, a scoping review methodology allows for a broad exploration of the existing literature, encompassing various study designs, methodologies, and geographic locations. Given the diverse backgrounds and experiences of international students worldwide, this approach is useful for capturing the breadth and depth of research on their mental health. Second, by adopting a scoping review approach, we seek to identify gaps and limitations in the existing literature systematically, which is necessary for feasibility evaluation of more targeted studies that address specific aspects of international student mental well-being and eventually the development of evidence-based interventions tailored to the unique needs of international students. To guide this broad exploration of literature and achieve the goals outlined above, the following objective and research questions have been established.

## Objective and review questions

The objective of this scoping review is to map and summarize research on international students' experiences and concerns (e.g., depression), as well as factors influencing their well-being (e.g., social support and institutional resources). The research questions below have been informed by the objective of the review, as well as guidance from the Joanna Briggs Institute (JBI) Manual of Evidence Synthesis [18]:

1. What mental health and well-being experiences and concerns do students report?

2. What individual, interpersonal, and environmental factors (e.g., acculturation stress, language barriers, availability of mental health services) are associated with the mental health and well-being of international students, and how do these factors vary across different regions or demographics?

## Eligibility criteria

The population, concept, and context (PCC) framework proposed by the JBI Manual for Evidence Synthesis [18] informed the inclusion and exclusion criteria (Table 1) of this scoping review.

**Table 1. Inclusion and exclusion criteria.**

| Criterion | Inclusion Criteria | Exclusion Criteria |
|---|---|---|
| Population | • International students studying abroad on temporary visa or refugee status.<br>• Studies including international and domestic students, if groups are clearly distinguished.<br>• No restrictions on student nationality, citizenship, age, gender, sexuality, or level of study. | • Studies focusing on short-term exchange students (typically <1 year).<br>• Studies that do not distinguish between international students studying abroad and those who returned to their home country. |
| Concept | • Studies focusing on mental health, well-being, mental health conditions (e.g., depression), mental health disorders (e.g., MDD), psychological adjustment, or acculturation stress.<br>• Mental health may be a primary or clearly defined secondary outcome. | • Studies focusing solely on health behaviors (e.g., alcohol use, sleep) without an explicit link to mental health or well-being. |
| Context | • Higher education institutions globally, including universities, colleges, and professional schools offering full-time academic degrees or certifications.<br>• No restrictions by geographic location. | • Non-traditional higher education settings (e.g., language schools, vocational schools, adult education centers, online platforms such as Coursera, Udemy). |
| Types of sources of evidence | • Peer-reviewed primary research studies published in English or French.<br>• No restriction on publication year. | • Grey literature (e.g., conference presentations), secondary sources (e.g., meta-analyses, systematic reviews). |

## Population

The population of focus is international students, including undergraduate and graduate levels, across various academic disciplines. International students, for the purpose of this scoping review, refer to individuals enrolled in higher education institutions outside of their home country, typically on a temporary visa or refugee status [19].

In terms of inclusion, synonyms such as "global", "foreign", "overseas", "abroad", and "cross-cultural" will be considered to ensure a comprehensive exploration of the diverse experiences of international students. Studies that include both international students and domestic students will be considered, if there is a clear distinction between the two populations (i.e., studies that draw a comparison between international students and domestic students).

For the exclusion criteria, studies that focus on international student populations enrolled in short-term exchange studies abroad (typically lasting less than one year) will be excluded, as there is not enough time to address the transition period and its effects on mental health in such programs [34]. Studies that do not differentiate samples of international students studying abroad versus those returning to their home countries in their analysis will also be excluded.

The review will not be restricted by student nationality, citizenship, age, gender, sexuality, or level of study, to allow for a broad understanding of the mental health concerns and needs within this dynamic population.

## Concept

Mental health encompasses a person's emotional, psychological, and social well-being, influencing how individuals think, feel, and behave [20]. Importantly, mental health is not merely the absence of mental disorders but also includes the ability to manage life's challenges, build healthy relationships, and achieve personal goals.

In this scoping review we will focus on the dimensions of "mental health" and "well-being", as well as mental health conditions and mental health disorders. Mental health conditions are a broad term that includes a range of emotional and psychological challenges that may affect well-being, including anxiety, depression, and stress. Meanwhile, mental health disorders are clinically diagnosed conditions that significantly impair cognitive, emotional, or social abilities, such as generalized anxiety disorder (GAD) and major depressive disorder (MDD) [21].

Given the broad definition of mental health, our inclusion criteria will include studies that consider concerns or experiences related to psychological well-being, emotional well-being, stress levels (including acculturation stress), and mental health status (e.g., depression). Studies examining "adjustment" in international students will also be included since this term is closely associated with acculturation stress and reflects how students cope with challenges of studying abroad. To ensure a comprehensive understanding of international student mental health, we will not exclude studies based on specific mental health status and disorders or their severity.

Studies that do not solely focus on mental health or well-being but include it as one of the clearly outlined components will also be included. For instance, studies that explore both student mental health and academic performance will be considered eligible. Additionally, if a study primarily focuses on physical health but includes a clear mental health component (e.g., reduction in stress levels), it will also be included in the review.

In terms of exclusion criteria, studies that focus solely on health behaviors that may be related to mental health, such as alcohol use or sleep patterns, without any explicit link to mental health or well-being will not be considered in this review. This exclusion ensures that we do not include studies that may skew our findings by making assumptions about the impact of these behaviors on mental health without direct evidence.

## Context

The context for this scoping review will center around higher education institutions globally. Higher education in this context includes various learning institutions that award academic degrees or professional certifications, such as universities,

colleges, and various professional schools in disciplines such as law, medicine, and technology where international students can pursue full-time studies [3].

With regards to inclusion criteria, the review will not be restricted by geographical location of the higher education institutions to capture diverse cultural contexts and experiences. This approach will allow us to identify common challenges across different geographical locations, which will enhance relevance and impact of our findings.

For the exclusion criteria, studies that mention "college" but do not pertain to a higher education context will be excluded upon full-text review. Additionally, non-traditional higher education settings such as language schools, vocational schools, adult education centers, and online learning platforms (e.g., Coursera, Udemy, Khan Academy) will be excluded, as they may have different academic structures and student demographics to the focus of this scoping review.

Any study that does not meet the eligibility criteria outlined above will be excluded from the scoping review.

### Types of sources of evidence

Additional to the PCC framework [18], this scoping review has other eligibility criteria. First, only materials available in English and French will be included in the review as both reviewers are proficient in these languages. Second, no year limit is applied to capture the historical perspectives as well as the recent developments in international student mental health. Third, secondary sources (e.g., meta-analysis and systematic reviews) and grey literature (e.g., conference presentations) will be excluded from the review. In cases of secondary sources meeting the PCC inclusion criteria [18], the sources will be retained, and the studies reviewed in them will be screened. Any study that does not meet the eligibility criteria outlined above will be excluded from the scoping review.

### Methods

The proposed scoping review (registered in OSF: 10.17605/OSF.IO/KH9DC) will be conducted in accordance with the JBI Manual for Evidence Synthesis [18], recommendations of Levac et al. [22] and will be reported in accordance with PRISMA-ScR guidelines [23] and PRISMA-P checklist (S1 File) [24] (https://www.prisma-statement.org/scoping).

### Sources of evidence

Four electronic databases (i.e., MEDLINE, Embase, and PsycInfo through Ovid, and CINAHL through EBSCO) were examined using distinct search strategies, to ensure thoroughness and comprehensiveness in addressing the proposed research topic. MEDLINE and Embase were selected because of their extensive coverage of biomedical literature, providing access to clinical studies relevant to mental health concerns experienced by international students in higher education contexts. PsycInfo was selected for its focus on psychology and behavioral sciences literature, offering insights into the cultural and social factors that influence international student mental health. Finally, CINAHL was selected for its focus on nursing and allied health literature, which includes research on the mental health of healthcare university students and various factors, including institutional factors, that influence their mental well-being, thus providing valuable insights applicable to understanding similar factors affecting international students.

To identify additional relevant studies, the reference lists of primary and secondary sources that meet the inclusion criteria will also be examined.

### Search strategy

The first author collaboratively developed the search strategy with guidance and input from the senior author and the Université de Montréal librarian. The senior author and the first author incorporated the scoping review objectives and PCC framework [18] into the initial search strategy. The librarian then provided input on refining the strategy, leading to further refinements based on a detailed examination of preliminary search results.

As the first step of the search strategy development, a preliminary search on the database MEDLINE was conducted. Following this, index terms used to categorize publications (i.e., descriptors), along with key words pertaining to the publication abstract, title and headings were identified and selected. The selected index terms and key words informed the development of a complete search strategy for MEDLINE (S2 File).

This search strategy was adapted for the three other electronic databases (i.e., Embase, PsycInfo, and CINAHL). While the MEDLINE search equation was maintained for Embase and PsycInfo, different database-specific descriptors were used to ensure a comprehensive search. For example, MEDLINE may use "exp Students," descriptor while Embase requires a broader set of descriptors, such as "student/ or college student/ or graduate student/ or health student/ or PhD student/ or postgraduate student/ or research student/ or social work student/ or student athlete/ or undergraduate student/ or university student/ or veterinary student/."

Additionally, the MEDLINE search equation was adjusted for CINAHL, accounting for differences such as "adj" versus "N" and other variations in search syntax.

The final searches for MEDLINE, Embase, and PsycInfo were conducted on March 27, 2024; the CINAHL search was conducted on April 4, 2024.

In addition to the database search, a backward reference search (i.e., screening the reference lists) of key primary and secondary sources will be conducted to identify studies that may not have been captured through the primary search strategy.

## Study selection

The study selection of articles appropriate for the proposed scoping review will be guided by the inclusion and exclusion criteria outlined above. The citations obtained from the databases search will be initially imported into EndNote [25] for manual duplicate and grey literature removal by the first author. Following this, the citations will be imported into Covidence, a web-based collaborative software platform commonly used to assist in knowledge synthesis [26], where any remaining duplicates will be automatically removed.

The study selection will first be piloted to ensure clarity of the eligibility criteria [18]. The pilot will be conducted in Microsoft Excel and will involve independent screening of subsets of 10 citations by two reviewers (first author and a member of the research team). Any discrepancies in the inclusion of studies will be resolved through discussion and consensus between the reviewers, and if needed, a third reviewer (senior author) will be consulted. Pilot testing will continue in subsets of 10 citations until a cumulative 90% agreement rate between the reviewers is achieved. The study selection process will then commence. To ensure consistency, an additional three subsets will be reviewed to confirm the continued 90% agreement rate.

The study selection process will proceed in Covidence and will involve two main stages: first, title and abstract screening, followed by full-text review. During title and abstract screening, studies will be assessed based on the eligibility criteria to determine their relevance to the research question. Subsequently, full-text review will involve a more detailed examination of the selected studies to confirm their eligibility for inclusion in the scoping review. Abstracts that do not contain sufficient information to assess eligibility will undergo full-text review.

Following the piloting process, only one reviewer (first author) will complete the screening process at each stage. In cases where there is uncertainty about the eligibility of an article, the first author will consult the second reviewer. If a consensus cannot be reached, a third reviewer (senior author) will be consulted to make the final decision, ensuring thoroughness and impartiality in the selection process.

## Data extraction

Following the conclusion of the study selection, data extraction process will begin. First, a charting tool will be developed to facilitate the extraction process and ensure its validity. The charting tool will outline categories of information to be extracted from the included studies and will be organized in Covidence.

Initially, the legend categories will be informed by the data categories listed in the JBI Manual of Evidence Synthesis [18], such as: title, Digital Object Identifier (DOI), author(s), year of publication, country of origin, sample size, study design, objective(s), analysis type, and key findings. Additionally, the categories within the charting tool will be informed by the research questions and the eligibility criteria of the scoping review, guided by the PCC framework [18]. The data extraction tool will be finalized upon a detailed review of 10% of the studies to identify additional relevant categories of information to extract. Moreover, to accommodate unexpected themes of information that may arise, open-ended categories will be added, optimizing comprehensive coverage of all relevant data.

Population-related data will include sample characteristics, such as sociodemographic characteristics (e.g., age, gender, country of origin, socioeconomic status) and levels (e.g., undergraduate, graduate studies) and focus of studies (e.g., nursing, law).

Concept-related data will encompass a wide range of mental health experiences. This includes identifying mental health concerns (e.g., depression) and positive experiences (e.g., life satisfaction), along with negative factors affecting mental health (e.g., social isolation) and positive factors contributing to mental well-being (e.g., institutional support).

Quantitative data will include numerical metrics including prevalence rates and scores on mental health scales. For example, if a study reports that 40% of students experienced symptoms of anxiety, this statistic will be extracted.

Qualitative data will focus on key themes or observations related to mental health status and contributing factors. For example, if a study highlights that international students described "stress due to visa challenges" as a common concern or reported "feeling supported by a peer mentorship program," these themes will be extracted as they reflect relevant factors.

In the case of longitudinal studies, data will be extracted at each time point reported in the study (e.g., at the start, midway, and completion) to ensure that changes over time are captured. For quantitative studies, numerical data (e.g., prevalence rates) will be collected at each time point. For qualitative studies, evolving factors affecting mental health will be identified, and observations or themes reported by the authors will be extracted separately for each respective time point.

Context-related data will include information pertaining to higher education setting, such as geographical location (e.g., country and city where the educational institution is located, rural or urban setting), and institutional characteristics (e.g., name of the institution, size of the institution).

The data extraction tool will remain adaptable, subject to iterative refinement in response to emergent findings. For incomplete or unclear information, assumptions will be made where appropriate and in consultation with the senior author. These assumptions will be specified in the report, so that it is clear to the reader where any assumptions were made. If clarification is not possible, only the data explicitly reported in the publication will be extracted.

Consistent with scoping review methodology [18] and the PRISMA-ScR guidelines [23], no formal risk of bias or certainty appraisal (e.g., GRADE assessment [27]) of the included studies will be conducted, as the purpose of this review is to map the available literature rather than assess individual study rigor. The extracted study design information will be presented to help the readers interpret the evidence within its methodological context.

The data charting table will be piloted on a random sample of included studies to ensure that the tool is comprehensive and easy to use [18,22]. Two reviewers (first author and a member of the research team) will be involved in the charting table pilot of 10 included studies. Both reviewers will independently extract the data and compare each other's results. They will then meet to discuss any discrepancies to reach a consensus. Once the pilot is successful, one reviewer (first author) will proceed with data extraction independently.

If needed, the first author will consult the other reviewer to resolve any uncertainties or discrepancies during the extraction process.

## Data synthesis and presentation

The extracted data will be summarized and presented using descriptive statistics (e.g., frequencies and percentages). Tables will be used to visually represent an overview of international student mental health.

A first table will incorporate sociodemographic information extracted from the studies. This table may include details such as age, gender, country of origin, level of education, and field of study.

The second table will include data on types of mental health concerns and positive experiences reported by international students (e.g., depression, life satisfaction). Quantitative data (e.g., prevalence rates) and qualitative data (e.g., themes such as "stress due to visa challenges" or "feeling supported by peer mentorship programs") will be presented. For qualitative data, the frequency with which each mental health concern (e.g., depression, anxiety) is mentioned across studies will be counted. For example, if five studies report depression as a concern, it will be noted as a recurring issue in the data. Content analysis will be used to organize multiple types of concerns into categories. Relevant quotes from studies will be included to illustrate the lived experiences of students.

Finally, a third table will include negative and positive factors reported by students that affect their mental health and well-being (e.g., lack of social support). For qualitative data on factors affecting mental health, the frequency with which specific factors are mentioned (e.g., lack of social support, academic pressure) across studies will be organized into categories and counted. Illustrative quotes from included studies will be provided to give context and depth to the reported factors, capturing students' experiences and perspectives.

This scoping review will not conduct statistical calculations or analyses to examine interactions between individual factors, as scoping reviews are not typically intended for such analyses (JBI Manual of Evidence Synthesis, 2020, Section 10.2.8) [18]. However, during the data extraction process, any patterns or associations that emerge – for example, if anxiety is more frequently reported by women or if students from a particular country show higher rates of depression – will be identified and reported.

Additional findings may be reported based on identification of patterns among the results obtained and their relevance to the objectives of this scoping review. For instance, if a distinct pattern emerges related to differences between domestic and international students or utilization of university support resources, such findings will warrant their own dedicated tables.

Moreover, additional forms of visual presentation might be needed to convey complex patterns or relationships identified in this scoping review, such as charts and graphs. These visual aids will enhance the understanding of the data.

The data underlying the findings will be made fully available without restriction within the final report or as supplementary files.

## Limitations

To our knowledge, this scoping review will be one of the first to summarize and comprehensively report on mental health of international students in higher education. However, this scoping review is not without its limitations.

First, it is important to highlight that extensive efforts were made to include comprehensive key terms in the search strategy, related to mental health and well-being, mental illness, as well as more specific mental health conditions, such as stress, anxiety, and depression. However, other terms and concepts related to mental health conditions (e.g., schizophrenia, bipolar disorder) were not included in the search strategy, as the intention is to focus on the most common and prevalent mental health concerns among international students, aiming to balance comprehensiveness with feasibility. As a result, there is a possibility that studies focusing on mental health conditions beyond those explicitly mentioned in the search strategy may be inadvertently excluded in this scoping review.

Second, although four databases – MEDLINE, Embase, PsycInfo, and CINAHL – were selected for a range of health research literature on the topic, additional databases, such as region-based, or databases pertinent to psychoeducational literature, including ERIC and Web of Science, could have been considered. As a result, there is a possibility that relevant studies within the psychoeducational domain can be overlooked.

Third, grey literature and secondary sources, including unpublished studies and conference abstracts, will not be considered in this scoping review. This decision was made due to concerns about the validity of grey literature, as these sources are not

peer-reviewed and may include preliminary results that could differ from the final published results. Consequently, there is a possibility that relevant studies might go undetected and not be included in the scoping review, thus leading to an incomplete representation of literature landscape. Additionally, since the review will only include published studies in English and French, there is a risk of publication bias towards western countries, which may skew the overall conclusions from this scoping review. Concurrently, our preliminary implementation of the search strategy identified studies conducted in non-western countries indicating some mitigation of this bias as studies conducted in non-western countries are also published in English or French.

Fourth, the scope of this review is limited to studies published in English and French. While this choice aims to be as inclusive as possible within the constraints of the reviewers' language proficiencies, it inevitably excludes research published in other languages. This language limitation may contribute to an overemphasis on certain geographic regions and student populations, reiterating the existing research biases and potentially overlooking relevant studies from non-English and non-French speaking contexts.

Fifth, there is a risk of bias in the selection process given that only one reviewer will be involved in the majority of that process (due to availability of resources). However, several strategies are in place to reduce this risk including piloting multiple subsets of 10 articles until a 90% agreement is reached, and continuing to pilot until three additional consecutive sets also results in a minimum of 90% agreement. Moreover, a randomization generator for selecting citations will be employed, ensuring a diverse representation of studies in the pilot.

Finally, this scoping review will not assess methodological rigour of the included studies – such as risk of bias or certainty appraisals. While such evaluation could provide valuable insights into the reliability and validity of collected evidence, the primary objective of this scoping review is to map the existing literature and identify key themes when it comes to mental health of international students. Information on study design will be extracted and presented; however, a lack of study quality assessment may reduce reliability of the findings, while at the same time acknowledging that the main purpose of scoping reviews is to map the literature rather than to assess its quality. As such, the impact of this scoping review may be limited in informing the selection of evidence-based interventions or in making specific recommendations for practice or policy in the field of international student mental health.

## Conclusion

Despite the outlined limitations, this scoping review aims to comprehensively map the available evidence on mental health among international students and highlight the gaps in current research. Through the synthesis of existing literature and key themes, this review will provide valuable insights into international student mental health experience. Additionally, it will identify the factors that both negatively and positively impact their well-being, offering a more nuanced understanding of the challenges and supports related to their mental health.

This scoping review will pave the way for future research to address these gaps and build upon the existing knowledge base, thus ensuring more inclusive and representative studies. This review seeks to contribute significantly to the field by uncovering areas for future research and informing policy and practice to better support the mental health needs of international students Furthermore, this synthesis of knowledge can provide a base and guidance for conducting a systematic review and meta-analysis on the topic of inquiry. Finally, the results of this scoping review may inform the development of targeted interventions and support services tailored to the diverse needs of the international student population.

## Supporting information

**S1 File. PRISMA-P Checklist.**
(DOCX)

**S2 File. Detailed search strategy for MEDLINE.**
(DOCX)

## Acknowledgments

The authors would like to acknowledge and thank Sarah Cherrier, a librarian at the Université de Montréal, for her invaluable assistance and support in refining the search strategy outlined in this protocol. Additionally, we extend our gratitude to Ayda Khalili, a research intern at the Youth Mental Health and Technology (YMHTech) Laboratory, for her input in validating the citations included in the protocol and Tania Sabatino, a Planning, Programming and Research Officer at the YMHTech Laboratory, for reviewing the protocol.

## Author contributions

**Conceptualization:** Kira Rudakova, Shalini Lal.

**Funding acquisition:** Shalini Lal.

**Investigation:** Kira Rudakova.

**Project administration:** Kira Rudakova, Shalini Lal.

**Resources:** Shalini Lal.

**Supervision:** Shalini Lal.

**Writing – original draft:** Kira Rudakova.

**Writing – review & editing:** Shalini Lal.

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
