## [Decision Letter · Decision Letter 0]

Dear Dr. Lal,

Thank you for submitting your manuscript to PLOS ONE. After careful consideration, we feel that it has merit but does not fully meet PLOS ONE’s publication criteria as it currently stands. Therefore, we invite you to submit a revised version of the manuscript that addresses the points raised during the review process.

Please find the attached file with some comments and suggestions. Some additions are highlighted in bold, while the marked sections indicate redundancies, areas where simplification may be possible, or points to reconsider.

We look forward to receiving your revised manuscript.

Kind regards,

Thiago P. Fernandes, PhD

Academic Editor

PLOS ONE

Reviewers' comments:

Reviewer's Responses to Questions

**Comments to the Author**

1. Does the manuscript provide a valid rationale for the proposed study, with clearly identified and justified research questions?

Reviewer #1: Yes

Reviewer #2: Yes

2. Is the protocol technically sound and planned in a manner that will lead to a meaningful outcome and allow testing the stated hypotheses?

Reviewer #1: Partly

Reviewer #2: Yes

3. Is the methodology feasible and described in sufficient detail to allow the work to be replicable?

Reviewer #1: Yes

Reviewer #2: Yes

4. Have the authors described where all data underlying the findings will be made available when the study is complete?

Reviewer #1: No

Reviewer #2: Yes

5. Is the manuscript presented in an intelligible fashion and written in standard English?

Reviewer #1: Yes

Reviewer #2: Yes

You may also provide optional suggestions and comments to authors that they might find helpful in planning their study.

Reviewer #1: The reviewed work, “Mental Health Concerns and Needs of International Students in Higher Education Settings: A Scoping Review Protocol,” addresses an important and timely issue regarding the mental health of international students in the context of higher education. The authors propose reviewing the existing literature on international students' mental health, which may provide valuable insights into their needs and challenges.

One of the work's strengths is the rigorous application of well-recognized methodological tools: the JBI Manual for Evidence Synthesis and the PRISMA-ScR and PRISMA-P guidelines. Using these tools ensures that the review will have a solid methodological foundation. Additionally, the carefully crafted search strategy, developed with the involvement of an experienced librarian, increases the likelihood that the search will be comprehensive and precise. The selection of four databases – MEDLINE, Embase, PsycINFO, and CINAHL – is justified, as each provides different but complementary perspectives on mental health, considering clinical, social, and healthcare-related factors.

Despite many advantages, certain limitations can be identified in the work, which may affect the overall quality and utility of the results. Firstly, the authors themselves note that their review will not include terms related to more severe mental disorders, such as schizophrenia or bipolar disorder. While their rationale for focusing on common mental health issues is reasonable, excluding other disorders may lead to an incomplete picture and skewed results.

A second significant limitation is the need for more methodological assessment of the included studies. The review will only map the literature in evaluating the quality of individual studies, which significantly limits the ability to draw firm conclusions regarding the effectiveness of interventions or recommendations for health policy. Relying solely on collected data without assessing quality may lead to less valuable findings for practitioners.

Additionally, excluding grey literature, such as conference abstracts or unpublished studies, could be problematic, especially in mental health, where some innovative research may not yet have been published in traditional journals. This could lead to publication bias and the omission of valuable data.

One of the most significant limitations of the reviewed work is the restriction to studies published in English and French. Mental health concerns of international students are a global issue, and excluding literature in other languages (e.g., Chinese, Spanish, Arabic) could lead to the omission of essential studies from regions where the number of international students is significant. Such a language choice may result in an overemphasis on Western models of mental health, limiting the ability to draw conclusions that are universal for different cultural contexts.

In conclusion, the proposed scoping review protocol is a well-structured and methodologically sound piece of work that can significantly contribute to the understanding of international students' mental health issues. However, certain limitations, such as the exclusion of more severe mental disorders, the lack of quality assessment of included studies, the omission of grey literature, and the language restriction, may weaken the strength and scope of the conclusions. Future studies could consider including a broader range of disorders, linguistic diversity, and a more rigorous assessment of study quality to provide a more comprehensive picture.

Reviewer #2: The scoping review plan is detailed and well argued for. I have a few minor questions:

1. Are you going to achieve an agreement rate with the data extraction when moving from two reviewers to one?

2. Expected contributions section, it isn’t clear from the introductory remarks that the current literature provides sufficient data to address ‘overlooked issues’? It would also help to specify, or provide examples, of what such overlooked issues might be.

3. The argument that “Therefore, a more comprehensive review that encompasses a diverse range of cultural backgrounds and host countries is needed for a holistic understanding of international student mental health concerns, needs, and experiences.” is sound, but can you provide any preliminary data or evidence to suggest that studies already exist to review that would support this endeavour?

4. How will you consider, if at all, longitudinal studies that examine this topic comparing pre vs post enrolment.

5. Will your analysis have the capacity to examine interactions between individual factors influencing student mental health?

**Do you want your identity to be public for this peer review?** For information about this choice, including consent withdrawal, please see our Privacy Policy

Reviewer #1: No

Reviewer #2: No

---

## [Author Response · Author response to Decision Letter 1]

16 Jan 2025

We have attached a Response to Reviewers File with detailed responses.

---

## [Decision Letter · Decision Letter 1]

Dear Dr. Lal,

Thank you for submitting your manuscript to PLOS ONE. After careful consideration, we feel that it has merit but does not fully meet PLOS ONE’s publication criteria as it currently stands. Therefore, we invite you to submit a revised version of the manuscript that addresses the points raised during the review process.

We look forward to receiving your revised manuscript.

Kind regards,

Thiago P. Fernandes, PhD

Academic Editor

PLOS ONE

Journal Requirements:

Reviewers' comments:

Reviewer's Responses to Questions

**Comments to the Author**

1. Does the manuscript provide a valid rationale for the proposed study, with clearly identified and justified research questions?

Reviewer #2: Yes

Reviewer #3: Yes

Reviewer #4: Yes

2. Is the protocol technically sound and planned in a manner that will lead to a meaningful outcome and allow testing the stated hypotheses?

Reviewer #2: Yes

Reviewer #3: Yes

Reviewer #4: Yes

3. Is the methodology feasible and described in sufficient detail to allow the work to be replicable?

Reviewer #2: Yes

Reviewer #3: Yes

Reviewer #4: Yes

4. Have the authors described where all data underlying the findings will be made available when the study is complete?

Reviewer #2: Yes

Reviewer #3: Yes

Reviewer #4: Yes

5. Is the manuscript presented in an intelligible fashion and written in standard English?

Reviewer #2: Yes

Reviewer #3: Yes

Reviewer #4: Yes

You may also provide optional suggestions and comments to authors that they might find helpful in planning their study.

Reviewer #2: The authors of the manuscript have addressed my previous comments, I don't have anything further to add.

Reviewer #3: I enjoy reading the rebuttal letter and the revised manuscript as the authors made assiduous efforts to address all the concerns raised by prior reviewers in a reasonable manner, and amended their manuscript accordingly. The current version of the manuscript is clearer and stronger. I only have relatively minor suggestions for the authors to consider as follows.

1. The Abstract, Under the Methods, “A search strategy guided by the Joanna Briggs Institute Manual (JBI) Manual of Evidence Synthesis”. The first “Manual” should be deleted.

2. P. 7, at the bottom, “”the Joanna Briggs Institute (JBI) manual of Evidence Synthesis [32]”. “manual” ought to be capitalized.

3. P. 7, last full paragraph, “The research questions below have been informed by the objective of the review, as well as the population, concept, and context (PCC) framework proposed by the Joanna Briggs Institute (JBI) manual of Evidence Synthesis [32]” vs. p. 8, under the Eligibility criteria, “The population, concept, and context (PCC) framework proposed by the JBI Manual for Evidence Synthesis [32]”. I feel the two writings are duplicated to a considerable extent. Furthermore, “the population, concept, and context (PCC)” had been expressed too many times throughout the manuscript.

Reviewer #4: Thank you for giving me the opportunity to review this manuscript.

-major

1) Multiple citations are listed without clearly linking them to specific claims, and some points are over-cited. Please align each argument with one or two appropriate references and avoid excessive citation.

2) Please explicitly describe the inclusion and exclusion criteria (e.g., age range, undergraduate/graduate status, study period, countries targeted).

3) There is an inconsistency: while the section mentions two reviewers, it later states that the first author alone will screen records. Please clarify exactly who performs each step (independently or jointly), and explain the role of automation tools like EndNote and Covidence. It is also better to involve at least two reviewers or include a cross-checking mechanism to ensure accuracy and reduce bias.

4) Please define any rules for handling different time points. Please state how the team will deal with incomplete or ambiguous data (e.g., contact authors, make assumptions).

5) Please note if a risk of bias assessment is omitted and explain why, in accordance with scoping review standards.

6) Please state that certainty assessments (like GRADE) are not conducted in this scoping review and provide a rationale.

-minor

1) The sentences of “The transition… signifies” were unnatural and lengthy. Please use concise expressions like, “Transitioning to higher education brings academic, personal, and emotional challenges.”

2) “What are individual, interpersonal, and environmental factors... are associated with...” should be correct to: “What individual, interpersonal, and environmental factors are associated with…” to fix the grammatical error.

3) While databases are listed, the date of last search for each source is not provided. Please indicate the search date for each database (e.g., “searched on March 5, 2024”) to ensure transparency.

I think it is necessary to revise the manuscript.

**Do you want your identity to be public for this peer review?** For information about this choice, including consent withdrawal, please see our Privacy Policy

Reviewer #2: No

Reviewer #3: No

Reviewer #4: No

---

## [Author Response · Author response to Decision Letter 2]

19 Jun 2025

We have uploaded a Response to Reviewers document that details our revisions.

---

## [Decision Letter · Decision Letter 2]

Mental health concerns and needs of international students in higher education settings: A scoping review protocol

PONE-D-24-34905R2

Dear Dr. Lal,

We’re pleased to inform you that your manuscript has been judged scientifically suitable for publication and will be formally accepted for publication once it meets all outstanding technical requirements.

Kind regards,

Thiago P. Fernandes, PhD

Academic Editor

PLOS ONE

Additional Editor Comments (optional):

Reviewers' comments:

Reviewer's Responses to Questions

**Comments to the Author**

1. Does the manuscript provide a valid rationale for the proposed study, with clearly identified and justified research questions?

Reviewer #4: Yes

2. Is the protocol technically sound and planned in a manner that will lead to a meaningful outcome and allow testing the stated hypotheses?

Reviewer #4: Yes

3. Is the methodology feasible and described in sufficient detail to allow the work to be replicable?

Reviewer #4: Yes

4. Have the authors described where all data underlying the findings will be made available when the study is complete?

Reviewer #4: Yes

5. Is the manuscript presented in an intelligible fashion and written in standard English?

Reviewer #4: Yes

You may also provide optional suggestions and comments to authors that they might find helpful in planning their study.

Reviewer #4: Thank you for revising the manuscript. I think this manuscript would be suitable for publication in this journal.

**Do you want your identity to be public for this peer review?** For information about this choice, including consent withdrawal, please see our Privacy Policy

Reviewer #4: No
